# Causal Explanations and XAI

**Sander Beckers**                                                                    SREKCEBREDNAS@GMAIL.COM
*University of Tübingen*

**Editors:** Bernhard Schölkopf, Caroline Uhler and Kun Zhang

## Abstract

Although standard Machine Learning models are optimized for making predictions about observations, more and more they are used for making predictions about the results of actions. An important goal of Explainable Artificial Intelligence (XAI) is to compensate for this mismatch by offering explanations about the predictions of an ML-model which ensure that they are reliably *action-guiding*. As action-guiding explanations are causal explanations, the literature on this topic is starting to embrace insights from the literature on causal models. Here I take a step further down this path by formally defining the causal notions of *sufficient explanations* and *counterfactual explanations*. I show how these notions relate to (and improve upon) existing work, and motivate their adequacy by illustrating how different explanations are action-guiding under different circumstances. Moreover, this work is the first to offer a formal definition of *actual causation* that is founded entirely in action-guiding explanations. Although the definitions are motivated by a focus on XAI, the analysis of causal explanation and actual causation applies in general. I also touch upon the significance of this work for fairness in AI by showing how actual causation can be used to improve the idea of path-specific counterfactual fairness.

**Keywords:** Explanation; Counterfactual; Actual Causation; Fairness

## 1. Introduction

Explainable Artificial Intelligence (XAI) is concerned with offering explanations of predictions made by Machine Learning models. Such predictions can serve many different purposes, each of which calls for a different kind of explanation. Part of the XAI literature is beginning to embrace the insight that Pearl's causal hierarchy is an invaluable tool to specify the requirements that different explanations need to fulfil for achieving those purposes (Pearl and Mackenzie, 2018). This hierarchy consists of a ladder that has three rungs: observations, interventions, and counterfactuals. Standard ML-models are optimized solely for making predictions about the first rung of the ladder, observations, and their widespread application is due to the fact that they are very successful in doing so. Yet there is also a growing interest in using ML-models for action-guidance, and that involves predictions about the second rung of the ladder, interventions (Lipton, 2018; Molnar et al., 2021; Schölkopf et al., 2021). Moreover, if the action occurs against the background of an already existing observation, then such predictions take place on the third rung of the ladder, counterfactuals.

Corresponding to these different purposes we have different notions of explanation that are appropriate for each. Simply put, purely associational measures – such as feature attribution (Ribeiro, 2016; Lundberg and Lee, 2017) – are appropriate when one requires explanations of the behavior of the ML-model itself: why did the model produce a particular output $Y = y$ for a particular input $I = i$? Such *model explanations* serve to better understand the model, be it to gain trust in the model or to perform model audit (Lipton, 2018; König et al., 2021).

Actions, on the other hand, take place in the real world, and thus acting on one feature might affect the value of another. Therefore if one desires explanations about the outputs produced as

the result of performing an action, then one is looking for *action-guiding explanations*. To find those one needs to go beyond the ML-model and consider the target system that it is a model of. Since action-guiding explanations are causal explanations, this requires knowledge of an additional model, namely a causal model of the target system. Fully specified causal models are notoriously hard to come by, but there are promising approaches that focus on combining partial causal models with ML-models (Mahajan et al., 2019; Karimi et al., 2020; Schölkopf et al., 2021). This paper focuses on a conceptual analysis of the relevant notions of causal explanations – as opposed to practical methods for obtaining such explanations – and therefore we assume perfect knowledge of a deterministic causal model of the target system. The results of weakening this assumption are to be investigated in future work.

Action-guiding explanations can occur in two basic forms. If one is interested in explaining under which conditions an action guarantees a particular output, then one needs **Sufficient Explanations**. If one is interested in finding an action that changes an observed input to produce a change in an already observed output, then **Counterfactual Explanations** are called for. [1] Beyond these basic forms, there exist many circumstances in which one is interested more generally in finding out which past actions actually explain an already observed output. Such explanations ask for the **Actual Causes** of an output, and it will turn out that they sit in between sufficient and counterfactual explanations in roughly the following way: an actual cause is a part of a good sufficient explanation for which there exist counterfactual values that would not have made the explanation better.

The next two sections provide the background context for our analysis. Sections 4 and 5 introduce various definitions of sufficient and counterfactual explanations by relating existing accounts of these notions to their causal counterparts. Section 6 offers a definition of actual causation, and concludes by presenting a novel definition of fairness based on actual causation.

## 2. Structural Equations Modeling

For the purposes of notational consistency across related work, this section reviews the definition of causal models along the lines of Halpern (2016) with little change.

**Definition 1** *A signature $\mathcal{S}$ is a tuple $(\mathcal{U}, \mathcal{V}, \mathcal{R})$, where $\mathcal{U}$ is a set of* exogenous *variables, $\mathcal{V}$ is a set of* endogenous *variables, and $\mathcal{R}$ a function that associates with every variable $Y \in \mathcal{U} \cup \mathcal{V}$ a nonempty set $\mathcal{R}(Y)$ of possible values for $Y$ (i.e., the set of values over which $Y$ ranges). If $\boldsymbol{X} = (X_1, \ldots, X_n)$, $\mathcal{R}(\boldsymbol{X})$ denotes the crossproduct $\mathcal{R}(X_1) \times \cdots \times \mathcal{R}(X_n)$.*

Exogenous variables represent factors whose causal origins are outside the scope of the causal model, such as background conditions and noise. The values of the endogenous variables, on the other hand, are causally determined by other variables within the model.

**Definition 2** *A causal model $M$ is a pair $(\mathcal{S}, \mathcal{F})$, where $\mathcal{S}$ is a signature and $\mathcal{F}$ defines a function that associates with each endogenous variable $X$ a* structural equation $F_X$ *giving the value of $X$ in terms of the values of other endogenous and exogenous variables. Formally, the equation $F_X$ maps $\mathcal{R}(\mathcal{U} \cup \mathcal{V} - \{X\})$ to $\mathcal{R}(X)$, so $F_X$ determines the value of $X$, given the values of all the other variables in $\mathcal{U} \cup \mathcal{V}$.*

We call a setting $\boldsymbol{u} \in \mathcal{R}(\mathcal{U})$ of values of exogenous variables a *context*. The value of $X$ may depend on the values of only a few other variables. $X$ *depends on $Y$ in context $\boldsymbol{u}$* if there is some

---

1. These fit within the broader category of explanations that focus on *necessity*, as opposed to *sufficiency*.

setting of the endogenous variables other than $X$ and $Y$ such that if the exogenous variables have value $u$, then varying the value of $Y$ in that context results in a variation in the value of $X$; that is, there is a setting $z$ of the endogenous variables other than $X$ and $Y$ and values $y$ and $y'$ of $Y$ such that $F_X(y, z, u) \neq F_X(y', z, u)$. We then say that $Y$ is a *parent* of $X$ and $X$ is a *child* of $Y$.

We extend this genealogical terminology in the usual manner, by taking the *ancestor* relation to be the transitive closure of the parent relation (i.e., $Y$ is an ancestor of $X$ iff there exist variables so that $Y$ is a parent of $V_1$, $V_1$ is a parent of $V_2$, ..., and $V_n$ is a parent of $X$). The *descendant* relation is simply the reversal of the ancestor relation (i.e., $X$ is a descendant of $Y$ iff $Y$ is an ancestor of $X$.) A *path* is a sequence of variables in which each element is a child of the previous element.

In this paper we restrict attention to *strongly recursive* models, that is, models where there is a partial order $\preceq$ on variables such that if $X$ depends on $Y$, then $Y \prec X$. In a strongly recursive model, given a context $u$, the values of all the remaining variables are determined (we can just solve for the value of the variables in the order given by $\preceq$).

An *intervention* has the form $X \leftarrow x$, where $X$ is a set of endogenous variables. Intuitively, this means that the values of the variables in $X$ are set to the values $x$. The structural equations define what happens in the presence of interventions. Setting the value of some variables $X$ to $x$ in a causal model $M = (\mathcal{S}, \mathcal{F})$ results in a new causal model, denoted $M_{X \leftarrow x}$, which is identical to $M$, except that $\mathcal{F}$ is replaced by $\mathcal{F}^{X \leftarrow x}$: for each variable $Y \notin X$, $F_Y^{X \leftarrow x} = F_Y$ (i.e., the equation for $Y$ is unchanged), while for each $X'$ in $X$, the equation $F_{X'}$ for $X'$ is replaced by $X' = x'$ (where $x'$ is the value in $x$ corresponding to $X'$).

Given a signature $\mathcal{S} = (\mathcal{U}, \mathcal{V}, \mathcal{R})$, an *atomic formula* is a formula of the form $X = x$, for $X \in \mathcal{V}$ and $x \in \mathcal{R}(X)$. A *causal formula (over $\mathcal{S}$)* is one of the form $[Y_1 \leftarrow y_1, \ldots, Y_k \leftarrow y_k]\phi$, where

- $\phi$ is a Boolean combination of atomic formulas,

- $Y_1, \ldots, Y_k$ are distinct variables in $\mathcal{V}$, and

- $y_i \in \mathcal{R}(Y_i)$ for each $1 \leq i \leq k$.

Such a formula is abbreviated as $[Y \leftarrow y]\phi$. The special case where $k = 0$ is abbreviated as $\phi$. Intuitively, $[Y_1 \leftarrow y_1, \ldots, Y_k \leftarrow y_k]\phi$ says that $\phi$ would hold if $Y_i$ were set to $y_i$, for $i = 1, \ldots, k$.

A causal formula $\psi$ is true or false in a *causal setting*, which is a causal model given a context. As usual, we write $(M, u) \models \psi$ if the causal formula $\psi$ is true in the causal setting $(M, u)$. (If $\psi$ is true for all contexts $u$ we write $M \models \psi$.) The $\models$ relation is defined inductively. $(M, u) \models X = x$ if the variable $X$ has value $x$ in the unique (since we are dealing with recursive models) solution to the equations in $M$ in context $u$ (i.e., the unique vector of values that simultaneously satisfies all equations in $M$ with the variables in $\mathcal{U}$ set to $u$). The truth of conjunctions and negations is defined in the standard way. Finally, $(M, u) \models [Y \leftarrow y]\phi$ if $(M_{Y \leftarrow y}, u) \models \phi$ (i.e., the intervention $Y \leftarrow y$ transforms $M$ into a new model $M_{Y \leftarrow y}$, in which we assess the truth of $\phi$).

## 3. Background

Assume that we have an ML-model which implements the overall function $h : \mathcal{R}(I) \rightarrow \mathcal{R}(Y)$, where $I$ consists of all the input variables and $Y$ is the single output variable. As mentioned, we also assume knowledge of a causal model $M$ of the target domain (i.e., that part of the world which $h$ is a model of). A first requirement is that such a causal model is consistent with the ML-model, meaning that they at least agree on how to classify all observations. A second requirement that

is helpful given the conceptual nature of our analysis is that the endogenous variables $\boldsymbol{I}$ suffice to obtain deterministic causal knowledge of the target domain. Technically this means that there are only two kinds of endogenous variables: those which are determined directly by the unobserved exogenous variables, and those which are determined entirely by members of $\boldsymbol{I}$. In future work both requirements can be loosened by adding probabilities to each. For the first, we can demand that both models are *likely* to agree on observations. For the second, we can add new exogenous variables and a probability distribution over them, which give us probabilistic causal models. These allow for probabilistic generalizations of all the definitions here presented.

**Definition 3** *A causal model* $M = ((\mathcal{V}, \mathcal{U}, \mathcal{R}), \mathcal{F})$ *agrees with* $h : \mathcal{R}(\boldsymbol{I}) \to \mathcal{R}(Y)$ *if* $\mathcal{V} = \boldsymbol{I} \cup \{Y\}$, *each* $V_i \in \mathcal{V}$ *either has a unique single exogenous parent* $U_i \in \mathcal{U}$ *or no exogenous parents, and for all* $\boldsymbol{i} \in \mathcal{R}(\boldsymbol{I})$, $y \in \mathcal{R}(Y)$: $M \models \boldsymbol{I} = \boldsymbol{i} \to Y = y$ *iff* $h(\boldsymbol{i}) = y$.

As knowledge of the causal model is hard to come by many approaches to action-guiding explanations ignore it entirely and assume that the input variables are causally independent (Sharma et al., 2020; Ustun et al., 2019; Poyiadzi et al., 2020; Kommiya Mothilal et al., 2021).

**Definition 4** *A causal model* $M$ *that agrees with* $h$ *satisfies* **Independence** *if for each* $I_a, I_b \in \boldsymbol{I}$, $I_a$ *is not a parent of* $I_b$.

Obviously **Independence** rarely holds, and thus any account of action-guiding explanation that depends on it is limited. Although this limitation is recognized (Mahajan et al., 2019; Karimi et al., 2021b; Molnar et al., 2021), this paper offers several results that formally characterize how extreme this limitation is. Simply put, under **Independence** a variety of very different causal notions become indistinguishable from each other. This holds for notions of sufficient explanation, counterfactual explanations, and various notions of actual causation.

Because of this limitation, work on action-guiding explanations in XAI has failed to take up the most relevant lesson that the literature on causation has to offer, namely that to give a causal explanation of an outcome is to give actual causes of that outcome (Woodward, 2003; Halpern and Pearl, 2005). An additional reason for this oversight is that the precise relation between explanation and prediction has been the subject of much debate in the history of philosophy of science.

Prediction in its most natural application is a forward-looking notion, meaning one predicts an event before it takes place. Explanation on the other hand is a backward-looking notion, meaning that one explains an event after it has happened. Yet as many papers on XAI clearly illustrate, explanations about past events are often required precisely to inform predictions about future events. Therefore a suitable notion of causal explanation, and thus also of actual causation, needs to specify how it relates to predictions. Given the tumultuous history that this relation has in the philosophy literature, it has been duly neglected in the philosophical work on causation, thereby obscuring the importance of causation for the practical goals that XAI is concerned with. (Hitchcock (2017) forms a notable exception!)[2] This paper corrects this by developing an account of causal explanation that shows both how it is connected to actual causation and how it can lead to action-guiding predictions.

Simply put, the goal of this paper is to upgrade the formalization of Woodward's influential philosophical account of causal explanation described below with the most recent insights from the causation literature, whilst also keeping track of the action-guiding demands that are prevalent in XAI (Woodward, 2003, p. 11).

---

2. In fact, his more informal exploration of the connection between actual causation and action-guiding prediction proceeds along somewhat similar lines as ours. One can view this paper as picking up where his left off.

Put differently, my idea is that one ought to be able to associate with any successful explanation a hypothetical or counterfactual experiment that shows us that and how manipulation of the factors mentioned in the explanation (the *explanans*, as philosophers call it) would be a way of manipulating or altering the phenomenon explained (the *explanandum*). Put in still another way, an explanation ought to be such that it can be used to answer what I call a *what-if-things-had-been-different question*: the explanation must enable us to see what sort of difference it would have made for the explanandum if the factors cited in the explanans had been different in various possible ways.

A crucial novel element of my account is the addition that a successful explanation must also be explicit about those factors that *may not be manipulated* for the explanation to hold, i.e., it must state which variables are to be safeguarded from interventions. Importantly, this is distinct from stating which variables must be held fixed at their actual values, for to hold variables fixed in fact means to intervene on them.

The following example (modified from (Karimi et al., 2021b)) is helpful for illustrating the various purposes that explanations can serve.

**Example 1** *Consider a system for loan applications that is captured by a causal model such that* $Y = (X_1 + 5 \cdot X_2 - 225,000) > 0$, *where* $Y$ *is a binary variable representing whether the loan is granted,* $X_1$ *is the applicant's income, and* $X_2$ *is the applicant's savings. Further, assume that the applicant's savings are determined by their initial deposit* $X_3$ *and their income in the following manner:* $X_2 = 3/10 \cdot X_1 + X_3$. *It is also the case that people with high savings take out a safety deposit box* ($X_4$) *at the bank:* $X_4 = X_2 > 1,000,000$.

Standard predictions in ML take place on the first rung of Pearl's causal ladder, namely that of observations. For example, an ML-model might pick up on the fact that all *observed* loan applicants who have a safety deposit box ($X_4 = 1$) also get a loan ($Y = 1$), and thus could predict that an applicant who has a safety deposit box will get a loan, i.e., it might learn a function so that $h(x_1, x_2, x_3, 1) = 1$ for all values $x_1, x_2, x_3$. In terms of the causal model, it is indeed the case that $M \models X_4 = 1 \rightarrow Y = 1$. Here $X_4 = 1$ is what Ribeiro et al. (2018) refer to as an *anchor*, which they interpret as a sufficient explanation of the outcome. Yet clearly such observational explanations are not a good guide towards action, for it would be absurd to recommend to someone to take out a safety deposit box so that their loan application is approved. This point can be brought out by making use of the second rung of the ladder, namely that of interventions: an applicant who takes out a safety deposit box as the result of being advised to do so will not necessarily get a loan: $M \not\models [X_4 \leftarrow 1]Y = 1$. In the same manner, interventions can be used to offer advice that *is* a good guide towards action, for example by telling an applicant that if they manage to obtain savings of $45,001$ then they are guaranteed to get a loan: $M \models [X_2 \leftarrow 45,001]Y = 1$.

## 4. Sufficient Explanations

**Sufficient Explanations** generalize the previous point by offering settings $\boldsymbol{X} = \boldsymbol{x}$ that causally suffice for $Y = y$. Existing approaches come in two flavors. Some approaches (Galhotra et al., 2021; Watson et al., 2021) define sufficient explanations using Pearl's notion of "probability of sufficiency", the deterministic version of which can be informally stated as: if we set the variables

in $X$ to $x$ and do not intervene on any of the other variables, then $Y$ takes on the values $y$.[3] Elsewhere I have called this interpretation of sufficiency *weak sufficiency* (Beckers, 2021).

**Definition 5**  $X = x$ *is weakly sufficient for* $Y = y$ *in* $M$ *if for all* $u \in \mathcal{R}(\mathcal{U})$ *we have that* $(M, u) \models [X \leftarrow x]Y = y$.

Other recent approaches take inspiration from logic and define sufficient explanations of an output as its *prime implicants*, giving so-called PI-explanations (Shih et al., 2018; Darwiche and Hirth, 2020; Koopman and Renooij, 2021). Informally: if we set the variables in $X$ to $x$, then $Y$ takes on the values $y$, regardless of the values of all other variables. In the context of causal models I have coined this *direct sufficiency* (Beckers, 2021).

**Definition 6**  $X = x$ *is directly sufficient for* $Y = y$ *in* $M$ *if for all* $c \in \mathcal{R}(C)$, *where* $C = \mathcal{V} - (X \cup Y)$, *and all* $u \in \mathcal{R}(\mathcal{U})$ *we have that* $(M, u) \models [X \leftarrow x, C \leftarrow c]Y = y$.

Obviously the second form of sufficient explanations is stronger. Coming back to our example, $X_2 = 45,001$ is a sufficient explanation for $Y = 1$ according to both readings. But $(X_1 = 50,000, X_3 = 25,000)$ is a sufficient explanation of $Y = 1$ only on the first reading, for although it is weakly sufficient for $Y = 1$ it is not directly sufficient. Which one of these notions is correct here? Imagine that a prospective applicant with these values is told that their income and initial deposit suffice for getting a loan. As a result the applicant concludes that there is no need to have such high savings and decides to spend $20,000$, so that their loan application is denied. The applicant would be quite right to be upset about this!

Such misunderstandings cannot occur when using direct sufficiency, as that gives us settings whose explanatory value is immune to the influence of interventions on other variables. Instead of concluding from this that one should always rely on direct sufficiency, I propose a generalization of sufficient explanations that adds an element to inform us explicitly as to which variables are assumed to be safeguarded from interventions. Concretely, in addition to specifying which variables need to be set to particular values, a sufficient explanation should also specify a set of variables $N$ that are *not* to be manipulated for the explanation to be action-guiding. Informally, if we set the variables in $X$ to $x$ and the variables in $N$ are safeguarded from interventions, then $Y$ takes on the value $y$, regardless of the values of all remaining variables.

I call the relevant notion of causal sufficiency at work *strong sufficiency* (Beckers, 2021), which can be formally defined as follows:

**Definition 7**  $X = x$ *is strongly sufficient for* $Y = y$ *in* $M$ *along* $N$ *if* $Y \subseteq N$ *and* $X = x$ *is directly sufficient for* $N = n$ *for some values* $n \supseteq y$.

Note that in this definition $N$ cannot just be any set, but rather we require that it is itself entirely determined by $X = x$. This is because the variables in $N$ can be thought of as a *network* that transmits the causal influence of $X$ to $Y$, and the idea of safeguarding this network is that it can continue fulfilling this role even when intervening on $X$. (I refer the reader to (Beckers, 2021) for an elaborate discussion of this definition as well as an equivalent alternative formulation.)

The following straightforward result shows the relative strengths of the above three notions of sufficiency.

---

3. This condition also appears in Halpern's notion of sufficient cause (Halpern, 2016).

**Proposition 8** *If $X = x$ is directly sufficient for $Y = y$ then $X = x$ is strongly sufficient for $Y = y$ along some $N$, and if $X = x$ is strongly sufficient for $Y = y$ along some $N$ then $X = x$ is weakly sufficient for $Y = y$.*

Using strong sufficiency we can define a sufficient explanation so that it specifies all the required elements for it to be action-guiding.

**Definition 9** *A pair $(X = x, N)$ is a **sufficient explanation** of $Y = y$ if $X = x$ is strongly sufficient for $Y = y$ along $N$. Any subset $X' = x'$ appearing in $X = x$ is called* a part of the explanation.
*If $(M, u) \models X = x$, we say that $(X = x, N)$ is an **actual sufficient explanation** of $Y = y$ in $(M, u)$.*
*If $N = Y$ we speak of a* direct sufficient explanation.

Of course we do not want to add redundant parts to a sufficient set, as a good explanation should be as concise as possible. Therefore a good sufficient explanation ought to be minimal with respect both to $X$ and to $N$.

**Definition 10** *A sufficient explanation $(X_1 = x_1, N_1)$ dominates an explanation $(X_2 = x_2, N_2)$ if both are explanations of the same $Y = y$, $X_1 \subseteq X_2$, and $N_1 \subseteq N_2$.*
*$(X_1 = x_1, N_1)$ strictly dominates $(X_2 = x_2, N_2)$ if $(X_1 = x_1, N_1)$ dominates $(X_2 = x_2, N_2)$ and not vice versa.*

This allows us to define what makes for a good sufficient explanation of an observed output.

**Definition 11** *An actual sufficient explanation $(X_1 = x_1, N_1)$ of $Y = y$ in $(M, u)$ is **good** if it is not dominated by any other actual sufficient explanation in $(M, u)$.*

To conclude the analysis of sufficient explanations, I offer a first result that shows the extreme limitation **Independence** poses. We have seen three very different notions of sufficiency, and yet under **Independence** they all collapse into one.

**Theorem 12** *If a causal model $M$ that agrees with $h$ satisfies **Independence** then the following statements are all equivalent:*

- *$X = x$ is weakly sufficient for $Y = y$ in $M$.*

- *$X = x$ is strongly sufficient for $Y = y$ in $M$.*

- *$X = x$ is directly sufficient for $Y = y$ in $M$.*

(Proofs of all Theorems are to be found in the Supplementary Material.)

## 5. Counterfactual Explanations

**Counterfactual Explanations** inform us which variables would have had to be different (and different in what way) for the outcome to be different. Although the XAI literature usually only considers one interpretation, it is important to distinguish between three different interpretations. The first interpretation is the standard one as it was introduced into the XAI literature (Wachter et al., 2017),

and is in fact appropriate only when one is looking either for model explanations or for a very restricted type of action-guiding explanations. An observation $I = i$ and output $Y = y$ is explained by offering an observation that is identical to the first one except for some variables $X$ that take on values $x'$ rather than $x$. The appropriate causal reading of this interpretation goes: if we had set the variables in $X$ to $x'$ rather than $x$ and held fixed all other variables at their actual values, then $Y$ would have been $y'$ rather than $y$. This is a deterministic version of what is usually called the direct effect (Pearl, 2001), which is action-guiding only if one can in fact hold fixed all other variables. Here I will call this relation *direct counterfactual dependence*.

The second interpretation is the usual one found in the causation literature, which takes into account that changing $X$ might change the values of other variables as well: if we had set the variables in $X$ to $x'$ rather than $x$ and had not intervened on any of the other variables, then $Y$ would have been $y'$ rather than $y$. This relation is standardly referred to as *counterfactual dependence* (Pearl, 2009).

The difference between these first two interpretations is gaining traction in recent work on algorithmic recourse (which is the term used for action-guiding given an already existing observation) (Karimi et al., 2021a). In fact, a slightly simpler version of Example 1 was initially used to illustrate this point precisely (Karimi et al., 2021b). Imagine an unsuccessful applicant with an income of $75,000$ and $25,000$ in savings. On the first interpretation, the "cheapest" counterfactual explanation would be that if their income had been $100,000$ rather than $75,000$ and everything else would have remain fixed, then they would have gotten the loan. The second interpretation takes into account the important fact that increases in income also result in increases in savings, and thus it would offer a cheaper counterfactual explanation: if their income had been $85,000$ rather than $75,000$ then they would have gotten the loan.

Does this mean that the second interpretation is the best one here? As before, an applicant hearing this explanation could be led to believe that they can ignore their savings, and thus spend some of it, which would of course invalidate the explanation given. Similar to the analysis of sufficient explanations, rather than choosing either one I contend instead that we should extend the explanation with information that ensures these misunderstandings cannot arise. This happens in two steps.

As a first step I introduce a third interpretation of counterfactual explanations, which is the explanatory counterpart to the most recent variant of Halpern & Pearl's influential definition of actual causation (the so-called modified definition) (Halpern, 2016). It generalizes the former two interpretations by including a "witness" to the explanation, meaning a set of variables $W$ so that if we had set the variables in $X$ to $x'$ rather than $x$, held fixed the variables in $W$ at their actual values, and had not intervened on any other variables, then $Y$ would have been $y'$ rather than $y$.[4]

The first interpretation above implicitly assumes that the witness contains all other variables, whereas the second interpretation assumes that the witness is empty. By spelling out the witness explicitly, the third interpretation includes the important information that the explanation remains valid only when holding fixed certain variables at their actual values. The first counterfactual explanation above, for example, depends for its validity on holding fixed $X_2$ (and is indifferent to the values of $X_3$ and $X_4$).

Formally the causal notion that this third interpretation relies on is the latest definition of actual causation by Halpern, which is best understood as a generalization of counterfactual dependence in

---

4. Note that in many contexts interventions on certain variables are feasible but come at a "cost", and thus one could quantify the "price" of explanations by building on the work of Karimi et al. (2021b).

the following way. (Throughout the rest of this paper, values $x'$ are assumed to differ from $x$ for *each* variable in $X$.)

**Definition 13** $Y = y$ counterfactually depends on $X = x$ rather than $X = x'$ in $(M, u)$ if $X$ is a *minimal set for which there exists some $W = w$ such that $(M, u) \models X = x \wedge W = w \wedge Y = y$ and $(M, u) \models [X \leftarrow x', W \leftarrow w]Y \neq y$.*

*We say that $W$ is a* witness. *If $W = \mathcal{V} \setminus (X \cup \{Y\})$, we speak of* direct counterfactual dependence. *If $W = \emptyset$, we speak of* standard counterfactual dependence. *(Note that in both cases we do not need to specify $W$.)*

The second step is to apply the insight we gained when analysing sufficient explanations, namely that an explanation should also mention the variables that are assumed to be safeguarded from interventions. This is precisely the information we need to avoid the above misunderstanding: it should be spelled out that the explanation assumes the applicant's savings will continue to be determined as they were. Therefore I define counterfactual explanations as counterfactual analogs of sufficient explanations.

**Definition 14** [5] *Given a causal setting $(M, u)$, if $((X = x, W = w), N)$ is an actual sufficient explanation of $Y = y$ and $((X = x', W = w), N)$ is a sufficient explanation of $Y = y'$ with $y' \neq y$ then we say that $X = x$ rather than $X = x'$ is a* **counterfactual explanation** *of $Y = y$ relative to $(W = w, N)$. We also write this as $(X = (x, x'), W = w, N)$.*

Just as with sufficient explanations, we want an explanation to be as concise as possible.

**Definition 15** *Given a causal setting $(M, u)$, we say that a counterfactual explanation $(X_1 = (x_1, x_1'), W_1 = w_1, N_1)$ dominates an explanation $(X_2 = (x_2, x_2'), W_2 = w_2, N_2)$ if both are explanations of the same $Y = y$, $X_1 \subseteq X_2$, $W_1 \subseteq W_2$, and $N_1 \subseteq N_2$.*

*As before,* strict domination *means that there is domination only in one direction. A counterfactual explanation is* **good** *if it is not dominated by any other counterfactual explanation.*

The following result shows that the previous two steps are indeed steps in the same direction:

**Theorem 16** *Given a causal setting $(M, u)$, the following two statements are equivalent:*

- *$Y = y$ counterfactually depends on $X = x$ rather than $X = x'$ in $(M, u)$.*

- *there exist $W_2$, $w_2 \in \mathcal{R}(W_2)$, and $N$ so that $(X = (x, x'), W_2 = w_2, N)$ is a good counterfactual explanation of $Y = y$.*

Coming back to our example, we can offer the cheaper of the two explanations and yet avoid any misunderstanding by saying that $X_1 = 75,000$ rather than $X_1 = 85,000$ is a good counterfactual explanation of $Y = 0$ relative to $(\{X_3 = 2,500\}, \{X_2\})$.

Similar to our result regarding the various notions of sufficiency, under **Independence** the different notions of counterfactual dependence collapse into one.

**Theorem 17** *If a causal model $M$ satisfies* **Independence** *then the following statements are all equivalent:*

---

5. Note that, unlike sufficient explanations, counterfactual explanations already come with a causal setting $(M, u)$ and there is no need to define an *actual* version of counterfactual explanations.

- $Y = y$ *directly counterfactually depends on* $\boldsymbol{X} = \boldsymbol{x}$ *rather than* $\boldsymbol{X} = \boldsymbol{x}'$ *in* $(M, \boldsymbol{u})$.

- $Y = y$ *standardly counterfactually depends on* $\boldsymbol{X} = \boldsymbol{x}$ *rather than* $\boldsymbol{X} = \boldsymbol{x}'$ *in* $(M, \boldsymbol{u})$.

- $Y = y$ *non-standardly and non-directly counterfactually depends on* $\boldsymbol{X} = \boldsymbol{x}$ *rather than* $\boldsymbol{X} = \boldsymbol{x}'$ *in* $(M, \boldsymbol{u})$, *(i.e., there is a witness* $\boldsymbol{W}$ *so that* $\emptyset \subset \boldsymbol{W} \subset \mathcal{V} \setminus (\boldsymbol{X} \cup \{Y\}))$.[6]

## 6. Actual Causation and Explanation

**Actual Causes** are those events which explain an output in the most general sense of explanation. There is a lively literature on how to define actual causation using causal models, but for the reasons outlined above, this literature has yet to find its way into the XAI literature. I here aim to set this straight by explicitly connecting actual causation to the notions of explanation that we have come across. A concise counterfactual explanation is good when you can get it, but often you cannot get it, and thus we need a weaker notion of explanation that is more generally applicable.[7] (The causal counterpart of this message is what initiated the formal causation literature some fifty years ago (Lewis, 1973).)

It is clear from the definitions that sufficient explanations are weaker than counterfactual explanations. But sufficient explanations ignore the counterfactual aspect entirely, which means they are of little value for action-guidance in the presence of an already existing observation. Therefore I define actual causes as parts of explanations that sit in between counterfactual and sufficient explanations: they are parts of good sufficient explanations such that there exist counterfactual values which would not have made the explanation better. This is weaker than demanding that the counterfactual values are part of a sufficient explanation of a *different* output, as we do for counterfactual explanations. Informally, $\boldsymbol{X} = \boldsymbol{x}$ rather than $\boldsymbol{X} = \boldsymbol{x}'$ is an *actual cause* of $Y = y$ if $\boldsymbol{X} = \boldsymbol{x}$ is part of a good sufficient explanation for $Y = y$ that could have not been made better by setting $\boldsymbol{X}$ to $\boldsymbol{x}'$.

First I make precise how changing the values of some variables can turn a sufficient explanation into a better one.

**Definition 18** *If* $((\boldsymbol{X} = \boldsymbol{x}, \boldsymbol{W} = \boldsymbol{w}), \boldsymbol{N})$ *is a sufficient explanation of* $Y = y$, *we say that* $\boldsymbol{X} = \boldsymbol{x}'$ *can replace* $\boldsymbol{X} = \boldsymbol{x}$ *if there exists a dominating explanation that includes* $(\boldsymbol{X} = \boldsymbol{x}', \boldsymbol{W} = \boldsymbol{w})$.

The following result ensures that the focus on $\boldsymbol{W}$ (instead of its subsets) in Definition 18 is without loss of generality.

**Proposition 19** *If* $((\boldsymbol{X} = \boldsymbol{x}, \boldsymbol{W} = \boldsymbol{w}), \boldsymbol{N})$ *is a sufficient explanation of* $Y = y$ *and there exists a dominating explanation* $((\boldsymbol{X} = \boldsymbol{x}', \boldsymbol{A} = \boldsymbol{a}), \boldsymbol{B})$ *for some values* $\boldsymbol{x}'$ *and* $\boldsymbol{a} \subseteq \boldsymbol{w}$, *then* $\boldsymbol{X} = \boldsymbol{x}'$ *can replace* $\boldsymbol{X} = \boldsymbol{x}$.

This allows us to formally define actual causation.

**Definition 20** $\boldsymbol{X} = \boldsymbol{x}$ *rather than* $\boldsymbol{X} = \boldsymbol{x}'$ *is an* **actual cause** *of* $Y = y$ *in* $(M, \boldsymbol{u})$ *if it is part of a good sufficient explanation of* $Y = y$ *in which* $\boldsymbol{X} = \boldsymbol{x}$ *cannot be replaced by* $\boldsymbol{X} = \boldsymbol{x}'$.

*If* $((\boldsymbol{X} = \boldsymbol{x}, \boldsymbol{W} = \boldsymbol{w}), \boldsymbol{N})$ *is the relevant good explanation, we say that* $\boldsymbol{X} = \boldsymbol{x}$ *rather than* $\boldsymbol{X} = \boldsymbol{x}'$ *is an actual cause of* $Y = y$ *relative to* $(\boldsymbol{W} = \boldsymbol{w}, \boldsymbol{N})$.

---

6. Obviously this holds only if $\mathcal{V} \setminus (\boldsymbol{X} \cup \{Y\})$ consists of at least two elements.

7. Obviously you can always find *some* counterfactual explanation of an output, for changing all of the variables will always allow you to get a different output. But an explanation that involves too many variables is of little use.

Except for a minor technical difference regarding the implementation of minimality, Definition 20 is equivalent to the definition of causation I developed in previous work (Beckers, 2021). Although I have here arrived at the same notion, I did so along a very different path, for in the previous work I did not draw any connection to explanations nor to action-guidance, but instead argued for the definition by contrasting it to other proposals for defining causation (including Definition 13).

Contrary to counterfactual explanations, actual causes do not guide you towards actions that, under the same conditions, would ensure the output to be different. But they do guide you towards actions that would not ensure the actual output under the same conditions as the actual action. For example, imagine a (very fortunate) applicant for whom $X_1 = 250,000$, $X_3 = 50,000$, and thus $X_2 = 125,000$. Obviously their application is successful, and their income by itself offers a good explanation of this fact: $X_1 = 250,000$ is a good direct sufficient explanation of $Y = 1$. Also, their income does not counterfactually explain the output, for the application would have been approved regardless of income. Yet we do have that $X_1 = 250,000$ rather than $X_1 = 200,000$ is an actual cause of the output, because we cannot replace $X_1 = 250,000$ by $X_1 = 200,000$ in our sufficient explanation (i.e., $X_1 = 200,000$ is not a sufficient explanation of $Y = 1$). This is helpful for example if the applicant is considering changing jobs and would like to know whether they can use up their savings and still get their application approved.

One might wonder whether we need a separate definition of actual causation, as opposed to simply considering all parts of good sufficient explanations to be causes. The distinction between these two options lies in the existence of alternative actions that would make the actual explanation worse, and it are precisely those actions which make actual causes good guides towards action. Consider for example a situation in which a short-circuit starts a fire ($F = 1$) in an office building. The flames set off the sprinklers ($S = 1$), and those put out the flames, preventing the building from burning down ($B = 0$) – where all variables are binary. Matching equations for this story are: $B = F \wedge \neg S$, $S = F$. In this scenario, the fire offers a good sufficient explanation of the building not burning down relative to the sprinklers functioning as they should. But obviously it would be unwise to conclude from this that we should start fires in order to prevent buildings from burning down! This point can be brought out by noting that the fire is *not* an actual cause of the building not burning down, because *there not being a fire* would have offered a better sufficient explanation of this outcome, as it does not rely on the sprinklers functioning properly. (Concretely: $F = 0$ can replace $F = 1$ in the good sufficient explanation ($F = 1, S$) of $B = 0$.) Since sprinklers can malfunction, or can be made to malfunction by a malicious actor, the best action is to not set fires in a building, in accordance with actual causation.

The following result specifies the claim that actual causes sit in between counterfactual and sufficient explanations: counterfactual explanations always contain actual causes (and obviously not vice versa).

**Theorem 21** *If $X_1 = x_1$ rather than $X_1 = x_1'$ is a counterfactual explanation of $Y = y$ in $(M, u)$ (relative to some $(W = w, N)$) then for some $X_2 \subseteq X_1$, $X_2 = x_2$ rather than $X_2 = x_2'$ is an actual cause of $Y = y$ in $(M, u)$ (where $x_2$ and $x_2'$ are the relevant restrictions to $X_2$).*

An obvious strengthening of actual causation is to replace the existential quantifier over counterfactual values with a universal one, so that the actual values are the *optimal* values in terms of explanations.

**Definition 22** $X = x$ is an **optimal cause** of $Y = y$ in $(M, u)$ if $X = x$ is part of a good sufficient explanation of $Y = y$ in which $X = x$ cannot be replaced. (I.e., there do not exist values $x'$ so that it could be replaced by those.)

Finally, replacing strong sufficiency with direct sufficiency offers a notion of *direct causation* .

**Definition 23** $X = x$ is a **direct cause** of $Y = y$ in $(M, u)$ if it is part of an actual direct good sufficient explanation of $Y = y$ in $(M, u)$.

**Proposition 24** [8] *If $X = x$ is a direct cause of $Y = y$ in $(M, u)$ then there exist values $x'$ such that $X = x$ rather than $X = x'$ is an actual cause of $Y = y$.*

We can now present a final result regarding the limitation of **Independence**: the different notions of causation also reduce to each other.

**Theorem 25** *If a causal model $M$ satisfies* **Independence** *then the following statements are all equivalent:*

- *$X = x$ is a direct cause of $Y = y$ in $(M, u)$.*

- *there exist values $x'$ so that $X = x$ rather than $X = x'$ is an actual cause of $Y = y$ in $(M, u)$.*

- *$X = x$ is part of a good sufficient explanation of $Y = y$ in $(M, u)$.*

### 6.1. Actual Causation and Fairness

So far this paper has focused on explanations as they relate to forward-looking action-guiding, but explanations (and actual causes in particular) are also of fundamental importance in backward-looking contexts, where the primary aim is to evaluate what has already happened. One such area that is of particular concern to XAI is that of fairness, which aims to evaluate whether a protected variable "contributed" in some way or other to produce an output. Here as well do we find a growing influence of causal models, most prominently in the work on counterfactual fairness, which in its basic form can be characterized as the condition that an output should not standardly counterfactually depend – see Def. 13 – on a protected variable (Kusner et al., 2017; Loftus et al., 2018). As we have seen, actual causes offer better explanations than counterfactual explanations, and thus I propose that fairness should be evaluated using actual causation instead.

Interestingly though, there already is a growing consensus that counterfactual dependence is too strong a condition and should be replaced with counterfactual dependence along *unfair paths* (again relying for its foundations on Pearl's work (Pearl, 2001)) (Loftus et al., 2018; Nabi and Shpitser, 2018; Chiappa, 2019; Wu et al., 2019). My definition of actual causation naturally accommodates this replacement as well, for actual causation occurs along a network $N$. Therefore we can define fairness by demanding that protected variables do not cause the outcome along an unfair network, i.e., a network that consists entirely of unfair paths.

Concretely, I propose the following deterministic definition of fairness, which can easily be extended into a probabilistic definition by moving to probabilistic causal models.

---

8. Direct causation does not imply optimal causation though. Here's a simple counterexample: $Y = (X = 1) \wedge A \vee X = 2$, with $\mathcal{R}(X) = \{0, 1, 2\}$ and $Y$ and $A$ binary. If we consider a context in which $A = 1$ and $X = 1$, then $X = 1$ is a direct cause of $Y = 1$ but not an optimal one.

**Definition 26** *Given a causal model $M$, a protected variable $A$, and a set of unfair paths $Pa_M$, we say that* the model is **fair** *for $A$ relative to $Pa_M$ if there does not exist a context $\boldsymbol{u}$ and values $y, a, a'$ so that $A = a$ rather than $A = a'$ is an actual cause of $Y = y$ in $(M, \boldsymbol{u})$ relative to some $(\boldsymbol{W} = \boldsymbol{w}, \boldsymbol{N})$ such that $Pa_{\boldsymbol{N}} \subseteq Pa_M$, where $Pa_{\boldsymbol{N}}$ are the paths in $\boldsymbol{N}$.*

We consider a very simple example as an illustration. The equations are $Y = \neg B \wedge \neg C$, $B = A$, $C = \neg A$, all variables are binary, and all paths from $A$ to $Y$ are unfair (meaning path-specific counterfactual fairness reduces to counterfactual fairness). $Y$ never standardly counterfactually depends on $A$, and thus existing notions of path-specific counterfactual fairness would always consider this model fair for $A$. In contrast, in a context where $A = 1$, $A = 1$ rather than $A = 0$ is an actual cause of $Y = 0$ (along $\{B\}$), and thus Definition 26 would consider this model unfair. Imagine $Y$ represents the outcome of a hiring process such that a candidate is hired if neither Billy ($B$) nor Cindy ($C$) rejects the applicant. Billy doesn't like religious people ($A = 1$), and therefore rejects their applications. Cindy doesn't like areligious people ($A = 0$), and therefore rejects their applications. So we get that a candidate's religiosity caused their application to be denied, and this matches the intuition that the outcome is unfair in a context where religiosity is a protected variable.

## 7. Conclusion

This paper has integrated work on causation into the field of action-guiding explainable AI by formally defining several causal notions of explanation as well as a definition of actual causation. These notions were motivated by demanding that an explanation contains all the elements required to ensure its validity and illustrating how they can be used as guides to action. I explored the connections between these notions as well as the consequences that come with ignoring the causal structure. Moreover, the proposed integration of actual causation with explanation extends beyond the practical needs of XAI to philosophy and science in general. This works offers a conceptual foundation that needs to be extended with probabilistic and approximate counterparts in future work. Lastly, the definition of actual causation was used to offer a novel account of causal fairness that improves upon the current state-of-the-art.

## Acknowledgments

I would like to thank the reviewers, as well as Timo Freiesleben, Konstantin Genin, Amir-Hossein Karimi, and Gunnar König for helpful comments on a preliminary version of this paper. This research was initially funded by the NIAS-Lorentz Theme-Group Fellowship on "Accountability in Medical Autonomous Expert Systems: Ethical and Epistemological Challenges for Explainable AI", and later by the Deutsche Forschungsgemeinschaft (DFG, German Research Foundation) under Germany's Excellence Strategy – EXC number 2064/1 – Project number 390727645

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

## Appendix A.

**Proposition 8** *If $X = x$ is directly sufficient for $Y = y$ then $X = x$ is strongly sufficient for $Y = y$ along some $N$, and if $X = x$ is strongly sufficient for $Y = y$ along some $N$ then $X = x$ is weakly sufficient for $Y = y$.*

**Proof:** Follows directly from the definitions. ∎

**Theorem 12** *If a causal model $M$ that agrees with $h$ satisfies* **Independence** *then the following statements are all equivalent:*

- $X = x$ *is weakly sufficient for $Y = y$ in $M$.*

- $X = x$ *is strongly sufficient for $Y = y$ in $M$.*

- $X = x$ *is directly sufficient for $Y = y$ in $M$.*

**Proof:** The implications from bottom to top are a direct consequence of Proposition 8.

Assume $X = x$ is weakly sufficient for $Y = y$ in $M$. This means that for all $u \in \mathcal{R}(\mathcal{U})$ we have that $(M, u) \models [X \leftarrow x]Y = y$. Let $C = \mathcal{V} - (X \cup \{Y\})$.

Given that $M$ agrees with $h$, either the equation for $Y$ is of the form $Y = U$ for some $U \in \mathcal{U}$, or $Y$ only has parents in $\mathcal{V} \setminus \{Y\}$. Since the former contradicts our assumption that in all contexts $(M, u) \models [X \leftarrow x]Y = y$, it has to be the latter.

As a consequence, interventions on all endogenous variables make the particular context irrelevant, i.e., for all $c \in \mathcal{R}(C)$ and all $u_1, u_2 \in \mathcal{R}(\mathcal{U})$, we have that $(M, u_1) \models [X \leftarrow x, C \leftarrow c]Y = y$ iff $(M, u_2) \models [X \leftarrow x, C \leftarrow c]Y = y$.

Further, for each $V_i \in \mathcal{V} \setminus \{Y\}$ the equation is of the form $V_i = U_i$. Although technically one could choose to define $\mathcal{R}(V_i)$ such that $\mathcal{R}(V_i) \not\subseteq \mathcal{R}(U_i)$, this comes down to defining a variable with values that it cannot obtain, which serves no purpose. Therefore we can assume that for each $c \in C$ there exists a context $u'$ such that $(M, u') \models C = c$. Given that no members of $X$ are ancestors of members of $C$, it also holds that $(M, u') \models [X \leftarrow x]C = c$. Since $X = x$ is weakly

sufficient for $Y = y$, we also have that $(M, \boldsymbol{u}') \models [\boldsymbol{X} \leftarrow \boldsymbol{x}]C = c \land Y = y$, from which it follows that $(M, \boldsymbol{u}') \models [\boldsymbol{X} \leftarrow \boldsymbol{x}, C \leftarrow c]Y = y$. Taken together with the previous observation that the particular context is irrelevant, we get that for all $\boldsymbol{c} \in \mathcal{R}(\mathcal{V} - (\boldsymbol{X} \cup \boldsymbol{Y}))$ and all $\boldsymbol{u} \in \mathcal{R}(\mathcal{U})$ we have that $(M, \boldsymbol{u}) \models [\boldsymbol{X} \leftarrow \boldsymbol{x}, C \leftarrow c]\boldsymbol{Y} = \boldsymbol{y}$, which is what we had to prove. ∎

**Theorem 16** *Given a causal setting $(M, \boldsymbol{u})$, the following two statements are equivalent:*

- *$Y = y$ counterfactually depends on $\boldsymbol{X} = \boldsymbol{x}$ rather than $\boldsymbol{X} = \boldsymbol{x}'$ in $(M, \boldsymbol{u})$.*

- *there exist $\boldsymbol{W_2}$, $\boldsymbol{w_2} \in \mathcal{R}(\boldsymbol{W_2})$, and $N$ so that $(\boldsymbol{X} = (\boldsymbol{x}, \boldsymbol{x}'), \boldsymbol{W_2} = \boldsymbol{w_2}, N)$ is a good counterfactual explanation of $Y = y$.*

**Proof:**

**Observation 1** *Recall from Definition 3 that exogenous variables only appear in equations of the form $V = U$. Say $\boldsymbol{R} \subseteq \mathcal{V}$ are all variables which have such an equation, and call these the root variables. It is clear that if we intervene on all of the root variables, they take over the role of the exogenous variables. Concretely, given strong recursivity, for any setting $\boldsymbol{r} \in \mathcal{R}(\boldsymbol{R})$ there exists a unique setting $\boldsymbol{v} \in \mathcal{R}(\mathcal{V})$ so that for all contexts $\boldsymbol{u} \in \mathcal{R}(\mathcal{U})$ we have that $(M, \boldsymbol{u}) \models [\boldsymbol{R} \leftarrow \boldsymbol{r}]\mathcal{V} = \boldsymbol{v}$.*

Assume that $Y = y$ counterfactually depends on $\boldsymbol{X} = \boldsymbol{x}$ rather than $\boldsymbol{X} = \boldsymbol{x}'$ in $(M, \boldsymbol{u})$ with witness $\boldsymbol{W_1}$, and $(M, \boldsymbol{u}) \models \boldsymbol{W_1} = \boldsymbol{w_1}$. This means that $(M, \boldsymbol{u}) \models \boldsymbol{X} = \boldsymbol{x} \land \boldsymbol{W_1} = \boldsymbol{w_1} \land Y = y$, and $(M, \boldsymbol{u}) \models [\boldsymbol{X} \leftarrow \boldsymbol{x}', \boldsymbol{W_1} \leftarrow \boldsymbol{w_1}]Y \neq y$.

Let $\boldsymbol{S} = \boldsymbol{R} \setminus (\boldsymbol{W_1} \cup \boldsymbol{X})$, and let $\boldsymbol{s} \in \mathcal{R}(S)$ be the unique values so that $(M, \boldsymbol{u}) \models \boldsymbol{S} = \boldsymbol{s}$.

As $\boldsymbol{R} \subseteq (\boldsymbol{S} \cup \boldsymbol{W_1} \cup \boldsymbol{X})$, we have that $(\boldsymbol{X} = \boldsymbol{x}, \boldsymbol{S} = \boldsymbol{s}, \boldsymbol{W_1} = \boldsymbol{w_1})$ is strongly sufficient for $Y = y$ along $N = \mathcal{V} \setminus (\boldsymbol{X} \cup \boldsymbol{W_1} \cup \boldsymbol{S})$, and thus $((\boldsymbol{X} = \boldsymbol{x}, \boldsymbol{S} = \boldsymbol{s}, \boldsymbol{W_1} = \boldsymbol{w_1}), N)$ is an actual sufficient explanation of $Y = y$.

Furthermore, changing $\boldsymbol{X}$ from $\boldsymbol{x}$ to $\boldsymbol{x}'$ obviously has no effect on any of the other values in $\boldsymbol{R}$. Therefore $(M, \boldsymbol{u}) \models [\boldsymbol{X} \leftarrow \boldsymbol{x}', \boldsymbol{W_1} \leftarrow \boldsymbol{w_1}]\boldsymbol{S} = \boldsymbol{s}$, and thus we get that $(M, \boldsymbol{u}) \models [\boldsymbol{X} \leftarrow \boldsymbol{x}', \boldsymbol{W_1} \leftarrow \boldsymbol{w_1}, \boldsymbol{S} \leftarrow \boldsymbol{s}]Y = y'$ for some $y' \neq y$. As before, we can conclude that $(\boldsymbol{X} = \boldsymbol{x}', \boldsymbol{S} = \boldsymbol{s}, \boldsymbol{W_1} = \boldsymbol{w_1})$ is strongly sufficient for $Y = y'$ along $N$, and thus $((\boldsymbol{X} = \boldsymbol{x}', \boldsymbol{S} = \boldsymbol{s}, \boldsymbol{W_1} = \boldsymbol{w_1}), N)$ is a sufficient explanation of $Y = y'$.

Combining the two previous paragraphs, we get that $(\boldsymbol{X} = (\boldsymbol{x}, \boldsymbol{x}'), (\boldsymbol{S} = \boldsymbol{s}, \boldsymbol{W_1} = \boldsymbol{w_1}), N)$ is a counterfactual explanation of $Y = y$. Let $(\boldsymbol{X} = (\boldsymbol{x}, \boldsymbol{x}'), \boldsymbol{W_2} = \boldsymbol{w_2}), N_2)$ be a dominating counterfactual explanation that is not dominated by any other explanation that contains $\boldsymbol{X} = (\boldsymbol{x}, \boldsymbol{x}')$. (It is easy to see that such an explanation must exist: one can simply keep removing elements from $\boldsymbol{W_1}$, $\boldsymbol{S}$, and $N$ until no further element can be removed while still remaining a counterfactual explanation of $Y = y$.)

Now assume that there exist $\boldsymbol{W_2}$, $\boldsymbol{w_2} \in \mathcal{R}(\boldsymbol{W_2})$, and $N$ so that $((\boldsymbol{X} = \boldsymbol{x}, \boldsymbol{W_2} = \boldsymbol{w_2}), N)$ is an actual sufficient explanation of $Y = y$ and $((\boldsymbol{X} = \boldsymbol{x}', \boldsymbol{W_2} = \boldsymbol{w_2}), N)$ is a sufficient explanation of some $Y = y'$ with $y' \neq y$. Since the first explanation is actual, it follows immediately that $(M, \boldsymbol{u}) \models \boldsymbol{X} = \boldsymbol{x} \land \boldsymbol{W_2} = \boldsymbol{w_2} \land Y = y$. Combining the second explanation with Proposition 8 we get that $(M, \boldsymbol{u}) \models [\boldsymbol{X} \leftarrow \boldsymbol{x}', \boldsymbol{W_2} \leftarrow \boldsymbol{w_2}]Y = y'$.

Note that we did not require $\boldsymbol{X}$ to be minimal in either direction, and thus the conditions as stated without minimality of $\boldsymbol{X}$ are equivalent. Therefore the conditions that include the minimality of $\boldsymbol{X}$ are also equivalent, which is what we had to prove. ∎

**Theorem 17** *If a causal model $M$ satisfies* **Independence** *then the following statements are all equivalent:*

- $Y = y$ *directly counterfactually depends on* $X = x$ *rather than* $X = x'$ *in* $(M, u)$.

- $Y = y$ *standardly counterfactually depends on* $X = x$ *rather than* $X = x'$ *in* $(M, u)$.

- $Y = y$ *non-standardly and non-directly counterfactually depends on* $X = x$ *rather than* $X = x'$ *in* $(M, u)$, *(i.e., there is a witness* $W$ *so that* $\emptyset \subset W \subset \mathcal{V} \setminus (X \cup \{Y\}))$.[9]

**Proof:** We show that we are free to choose the witness $W$ as we like, from which the result follows.

Assume $Y = y$ counterfactually depends on $X = x$ rather than $X = x'$ with witness $W_1$ and $(Mu) \models W_1 = w_1$ for some values $w_1$. Take any set $W_2 \subseteq (\mathcal{V} \setminus (X \cup \{Y\}))$, and let $w_2$ be the values of $W_2$ in $(M, u)$. First, note that we have $(M, u) \models X = x \wedge W_2 = w_2 \wedge Y = y$. Second, since none of the members of $X$ are ancestors of any of the members of $W_2$, we also have that $(M, u) \models [X \leftarrow x']W_2 = w_2 \wedge Y = y'$. Thus we also have that $(M, u) \models [X \leftarrow x', W_2 \leftarrow w_2]Y = y'$, and therefore also that $Y = y$ counterfactually depends on $X = x$ rather than $X = x'$ with witness $W_2$. ∎

**Proposition 19** *If* $((X = x, W = w), N)$ *is a sufficient explanation of* $Y = y$ *and there exists a dominating explanation* $((X = x', A = a), B)$ *for some values* $x'$ *and* $a \subseteq w$, *then* $X = x'$ *can replace* $X = x$.

**Proof:** Assume $((X = x, W = w), N)$ is a sufficient explanation of $Y = y$ and $((X = x', A = a), B)$ is a dominating explanation of $Y = y$, with $a \subseteq w$. We show that $((X = x', W = w), B)$ is a sufficient explanation of $Y = y$, from which the result follows.

Let $C = \mathcal{V} \setminus (X \cup A \cup B)$. From the definition of sufficient explanations, we know that for all $u \in \mathcal{R}(\mathcal{U})$ and all $c \in \mathcal{R}(C)$, we have that $(M, u) \models [X \leftarrow x', A \leftarrow a, C \leftarrow c]B = b$ for some $b \in \mathcal{R}(B)$ that includes $y$.

Let $D = \mathcal{V} \setminus (X \cup W \cup B)$, $F = W \setminus A$, and let $f$ be the restriction of $w$ to $F$. Note that $C = F \cup D$. From the previous paragraph it follows that for all $u \in \mathcal{R}(\mathcal{U})$ and all $d \in \mathcal{R}(D)$, we have that $(M, u) \models [X \leftarrow x', A \leftarrow a, F \leftarrow f, D \leftarrow d]B = b$, and thus $(M, u) \models [X \leftarrow x', W \leftarrow w, D \leftarrow d]B = b$, which is what had to be shown. ∎

**Theorem 21** *If* $X_1 = x_1$ *rather than* $X_1 = x_1'$ *is a counterfactual explanation of* $Y = y$ *in* $(M, u)$ *(relative to some* $(W = w, N)$*) then for some* $X_2 \subseteq X_1$, $X_2 = x_2$ *rather than* $X_2 = x_2'$ *is an actual cause of* $Y = y$ *in* $(M, u)$ *(where* $x_2$ *and* $x_2'$ *are the relevant restrictions to* $X_2$*).*

**Proof:** Assume $X_1 = x_1$ rather than $X_1 = x_1'$ is a counterfactual explanation of $Y = y$ in $(M, u)$ relative to $(W = w, N)$. This means that $((X_1 = x_1, W = w), N)$ is an actual sufficient explanation of $Y = y$ and $((X_1 = x_1', W = w), N)$ is a sufficient explanation of $Y = y'$ with $y' \neq y$.

Let $(T = t, S)$ be a good sufficient explanation of $Y = y$, i.e., an actual sufficient explanation of $Y = y$ that dominates $((X_1 = x_1, W = w), N)$ and cannot itself be dominated by another

---

9. Obviously this holds only if $\mathcal{V} \setminus (X \cup \{Y\})$ consists of at least two elements.

actual sufficient explanation of $Y = y$. (It is easy to see that such an explanation must exist: one can simply keep removing elements from $((X_1 = x_1, W = w), N)$ until no further element can be removed while still remaining a sufficient explanation of $Y = y$.)

Let $X_2 = X_1 \cap T$. We now show that $X_2 \neq \emptyset$ by a reductio.

Assume $X_2 = \emptyset$. This means that $T \subseteq W$. Also, $S \subseteq N$. Let $s \in \mathcal{R}(S)$ and $n \in \mathcal{R}(N)$ be the actual values of $S$ and $N$ in $(M, u)$.

Let $C = \mathcal{V} \setminus (X_1 \cup W \cup N)$. Given that $((X_1 = x_1', W = w), N)$ is a sufficient explanation of $Y = y'$, we have that for all $c \in \mathcal{R}(C)$, $(M, u) \models [X_1 \leftarrow x_1', W \leftarrow w, C \leftarrow c]Y = y'$.

Let $D = \mathcal{V} \setminus (T \cup S)$. Given that $(T = t, S)$ is a sufficient explanation of $Y = y$, we have that for all $d \in \mathcal{R}(D)$, $(M, u) \models [T \leftarrow t, D \leftarrow d]Y = y$.

By our assumption, $X_1 \subseteq (W \setminus T)$. Thus $D = X_1 \cup C \cup (W \setminus (T \cup X_1)) \cup (N \setminus S)$. Therefore from the previous paragraph we get that for all $c \in \mathcal{R}(C)$, $(M, u) \models [X_1 \leftarrow x_1', W \leftarrow w, C \leftarrow c]Y = y$. This contradicts the paragraph before the previous, and therefore $X_2 \neq \emptyset$.

Let $W_2 = T \setminus X_2$. Then we can conclude that $(X_2 = x_2, W_2 = w_2, S)$ is a good sufficient explanation of $Y = y$, where $x_2$ is the restriction of $x_1$ to $X_2$, and $w_2$ is the restriction of $W$ to $W_2$.

Remains to be shown that $x_2'$ cannot replace $x_2$ in this explanation, where $x_2'$ is the restriction of $x_1'$ to $X_2$. The former just means that $((X_2 = x_2', W_2 = w_2), S_2)$ is not a sufficient explanation of $Y = y$ for any $S_2 \subseteq S$.

Again we proceed by a reductio: assume that $((X_2 = x_2', W_2 = w_2), S_2)$ is a sufficient explanation of $Y = y$. Let $F = \mathcal{V} \setminus (X_2 \cup W_2 \cup S_2)$. We have that for all $f \in \mathcal{R}(F)$, $(M, u) \models [X_2 \leftarrow x_2', W_2 \leftarrow w_2, F \leftarrow f]Y = y$. In particular, we have that $(M, u) \models [X_1 \leftarrow x_1', W \leftarrow w]Y = y$.

Recall that $((X_1 = x_1', W = w), N)$ is a sufficient explanation of $Y = y'$ with $y' \neq y$. Using Proposition 8, we get that $(M, u) \models [X_1 \leftarrow x_1', W \leftarrow w]Y = y'$. This contradicts the result in the previous paragraph, which concludes the proof. ∎

**Proposition 24** *If $X = x$ is a direct cause of $Y = y$ in $(M, u)$ then there exist values $x'$ such that $X = x$ rather than $X = x'$ is an actual cause of $Y = y$.*

**Proof:** Assume $X = x$ is a direct cause of $Y = y$ in $(M, u)$, i.e., it is part of a direct good sufficient explanation $(X = x, W = w)$ of $Y = y$ in $(M, u)$. This means that all that remains to be shown, is that there exist values $x' \in \mathcal{R}(X)$ such that $(X = x', W = w)$ is not a direct sufficient explanation of $Y = y$.

We know that $(X = x, W = w)$ is directly sufficient for $Y = y$, and this does not hold if we remove any subset from either $X$ or $W$. Let $C = \mathcal{V} \setminus (X \cup W \cup \{Y\})$. Then we have that for all $c \in \mathcal{R}(C)$, $(M, u) \models [X \leftarrow x, W \leftarrow w, C \leftarrow c]Y = y$. From the minimality of $X$, it follows that there exists a $c \in \mathcal{R}(C)$ and $x' \in \mathcal{R}(X)$ so that $(M, u) \models [X \leftarrow x', W \leftarrow w, C \leftarrow c]Y \neq y$. Therefore $(X = x', W = w)$ is not a direct sufficient explanation of $Y = y$. ∎

**Theorem 25** *If a causal model $M$ satisfies* **Independence** *then the following statements are all equivalent:*

- $X = x$ *is a direct cause of* $Y = y$ *in* $(M, u)$.

- *there exist values $\boldsymbol{x}'$ so that $\boldsymbol{X} = \boldsymbol{x}$ rather than $\boldsymbol{X} = \boldsymbol{x}'$ is an actual cause of $Y = y$ in $(M, \boldsymbol{u})$.*

- $\boldsymbol{X} = \boldsymbol{x}$ *is part of a good sufficient explanation of $Y = y$ in $(M, \boldsymbol{u})$.*

**Proof:** The implication from the first statement to the second is a direct consequence of Proposition 24.

The implication from the second statement to the third follows from the definition of actual cause.

The implication from the third statement to the first follows from Theorem 12, which shows that under **Independence** we may replace sufficiency with direct sufficiency, and thus the result follows from the definition of direct cause. ∎

