# OpenReview forum: "Causal Explanations and XAI"
_cclear.cc/CLeaR/2022/Conference — CLeaR 2022 Poster_

### Official Review · Reviewer_77oH · 2021-11-22

**Confidence:** 3
**Overall Score:** 7

**Main Review:**

This paper introduces and motivates refined notions of causal explanations in deterministic structural causal models. These are compared with alternative notions of causal explanations and discussed alongside the corresponding notions of causation. It is being argued for why the presented notion of causal explanations is useful beyond the already existing notions. The rather abstract discussion is illuminated with a few examples. Overall a good paper, accept.



# Pros:

* Well written

# Cons:

* unclear what is new in this paper.

* significance for machine learning is mostly limited to this conceptual extension of causal terminoloy


# Originality

This point is my major concern. I find it not obvious to disentangle which parts of the paper are a collection and comparison of notions that have already appeared elsewhere and which parts are new to the paper. In my opinion the paper would benefit from stating this more clearly, for example by including a list of the key novel contributions in the introduction.
As far as I understand, the various notions of causation (weak sufficiency, direct sufficiency, strong sufficiency, counterfactual dependence, actual causation) are taken from the literature (mostly Beckers, 2021) while the notions of causal explanations derived thereof are new to this paper. This, i.e., going from notions of causation to notions of causal explanations, is a logical step to take. Further novel aspects are the proofs that several distinct notions of causation and explanation collapse under the independence assumption as well as the (brief) application to fairness.

# Significance

As showcased by the examples, the proposed notion of sufficient explanation does capture important information about how the actual value of the target variable has been obtained or how the desired value can be obtained. I can see this being relevant for action-guiding and thus think the discussion is relevant from a general point of view. The significance for machine learning is mostly limited to this conceptual extension of causal terminology, while it is not discussed how this new terminology can be implemented concretely to explain the decisions of machine learning models or how to inform machine learning models by causal knowledge. In this sense one might say that the paper is more about causal reasoning in general rather than about XAI, but this is a matter of perspective I suppose. Also, the demonstration of how restricting the independence assumption is as well as the discussion on fairness in section 6.1 appear relevant to XAI.


# Quality

The paper is of good quality. The discussion and comparison of several notions of what causation can be understood to mean in a structural causal model, with fine but important variations, as well as the associated notions of causal explanations appear quite illuminating to me. Moreover, the propositions and theorems answer questions that naturally come up when reading the definitions and discussion. While I did not check the proofs, the statements seem plausible. There are no obvious errors that I can see.

# Clarity

Due to the level of abstraction and the long list of interdependent definitions with fine variations, the paper is in my point of view a tough read. However, I think this is mostly due to the nature of the topic rather than the clarity of writing and exposition. In fact, the paper follows a clear thread and the author takes care to motivate the definitions and to illustrate some of them with examples. That being said, there are a few details in which I think the discussion could be more clear or precise, see the section "Other comments" below.



# Further comments

* First paragraph of section 3: The term "target domain" is somewhat vague, what exactly needs to be known?
* "Technically this means that there are only two kinds of endogenous variables..." in the first paragraph of section 3. Why is this required? Even if I_b depends on I_a and an endogeneous variable V(I_b), Y could still be deterministically determined by \bold{I}, could it not? I suppose you want that Y does not depend on an endogeneous variable which is not in \bold{I}.
* In Definition 3 the statement "I = i --> Y = y" is used as a causal formula. I suppose "\bold{I} = \bold{i} --> Y = y" means that if \bold{I} = \bold{i} then Y = y. However, I do not see how this fits into the general definition of causal formulas in section 2.
* What is the set C in definition 6? Is it C = V \ (X \cup Y)? Or is the statement supposed to hold for all or for one C \subset V \ (X \cup Y)?
* Typo in Theorem 16: There is an additional closing parenthesis in "W_2 = w_2)"
* The fire / burning house example confuses me because I do not see how it makes a case for using actual causation. Do you mean that F = 1 is not actual cause of B = 0 while F = 0 is?
* What does "causes" mean in Definition 25? I suppose this refers to actual causation, right?

**Summary:**

This paper introduces and motivates refined notions of causal explanations in deterministic structural causal models.

---

> ### Author Response · Authors · 2021-11-29
> **Response to review**
>
> Thank you for your overall positive feedback and careful comments!
>
> I am quite surprised though that the reviewer identifies lack of originality as their major problem with the paper, because it is the first to formally define distinct notions of causal explanations and relate them to actual causation, and it is certainly the first bridge the gap (the chasm!) between the literature on actual causation and XAI. (For sake of completeness, I should point out though that since writing the paper, I came across Chris Hitchcock’s paper “Actual causation: what’s the use?” which also addresses the connection between actual causation and action-guidance, and I will cite it. However, its discussion is entirely informal and without much detail, so it’s not a competitor to this paper.)
>
> The reviewer claims that going from the causal notions to the corresponding explanatory notions is a logical step to take, and I fully agree, which is what makes it all the more surprising that so far nobody had taken this step. (Beckers, for example, does not mention the word explanation once, nor does he talk about action-guiding. Halpern & Pearl do have a paper on explanations which uses their definition of causation, but it doesn’t offer distinct notions of explanation, nor does it connect to XAI in any way.) Moreover, over the past twenty years a huge literature has emerged discussing what definition of actual causation is the right one, with no consensus in sight. By connecting the definition of Beckers to explanations and action-guidance, this paper offers strong and novel support for that definition. This holds in particular because it is not at all clear if and how most other definitions of actual causation can be transformed into a corresponding explanatory notion.
>
> I agree with the reviewer that the novelty of these contributions may be lost if they are not mentioned more explicitly, and thus I will take up their suggestion of listing them in the introduction.
>
> Further comments:
> - The target domain here simply refers to that part of the world that is being described by the variables and their interaction. I will make this explicit.
> - I am not sure how the reviewer’s question relates to their comment. The idea is simply that the variables are nicely split up into two groups: those which are determined entirely by the other variables in I, and those which are not determined by any of the variables in I and are thus causal ancestors of the variables in I. The reason for this restriction is that the language of causal models (in the notation that I use) does not allow using exogenous variables in causal formulas, and thus we want to make sure that all variables about which we want to make statements are endogenous.
> - Causal formulas are said to be Boolean combinations of atomic formulas, and those include disjunctions. The implication is purely propositional here, and therefore it is equivalent to a disjunction, which is why I slightly abuse notation.
> - That was a bit sloppy… The set C is implicitly defined when mentioning c as belonging to the range of V - (X u Y). I will make this explicit.
> - Thanks for catching the typo.
> - Yes, that is indeed what I mean, and I should mention this explicitly. The structure of this example is a common one in the literature on actual causation.
> - Yes. Again being sloppy, as Definition 19 does not use that term… I will simply write actual cause instead.

---

### Official Review · Reviewer_u6p5 · 2021-11-24

**Confidence:** 4
**Overall Score:** 6

**Main Review:**

This paper introduce the relations between XAI and causation, and give some clearly and insteresting definination of causation based explaination. I think such discussion contributes to make the action guiding explaination to go one step further. BTW, is it necessary to require the ML model and the causal model agree on each other? Maybe it can be relaxed.

**Summary:**

This paper give discussion about sufficient explaination, counterfactual explaination, and actual causation explaination.

---

> ### Author Response · Authors · 2021-11-29
> **Response to review**
>
> Thanks for your positive comments! Regarding whether it is necessary that the ML-model and the causal model agree on all observations, note that the paper already addresses this to some extent, for it says: “In future work both requirements can be loosened by adding probabilities to each. For the first, we can demand that both models are *likely* to agree on observations.”
>
> Of course this short comment leaves much to be determined. For example, if the causal model disagrees with the ML-model for some input-output pair, do we want to assume that at least one of them is correct? And if so, should it always be the same model? Or should we generalize such that both models come with probabilities which represent their uncertainty? The latter scenario would be most natural if both models were learnt entirely independently of each other, but that seems unrealistic (and inefficient). In fact, given that ML-methods are influential precisely because they are so good at learning associations, it seems likely that we would use the function h as a constraint when learning the solutions to the causal model in the absence of interventions, and then — per construction — both models would agree on all observations. I believe addressing these issues (and how to incorporate uncertainty more generally) merits a paper of its own. The reason is that otherwise one runs the risk of conflating conceptual and definitional issues with practical empirical issues.

---

### Official Review · Reviewer_7Td3 · 2021-11-27

**Confidence:** 4
**Overall Score:** 7

**Main Review:**

The introduction distinguishes between explaining the model itself and explaining the real world. These are profoundly different things targets for explanation, and much of the work in XAI has tended to focus on the first. That is, work in XAI aims to answer the question: “Why did the model make conclusion X?” rather than “Why would making conclusion X be justified by the way that the world works?” While this is made fairly clear later in the paper, the beginning of the paper reads as if the authors believe that most work in XAI targets the latter (much more difficult) question. The introduction should be revised to better reflect the later material.

The focus of the paper (“…conceptual analysis of the relevant notions of causal explanations…”) is interesting, though limited. Still, it is a sufficient contribution to warrant acceptance.

The authors claim that “If one is interested in ﬁnding an action that changes an already observed output, then Counterfactual Explanations are called for.” This is not what many researchers mean when they say “counterfactual explanation”.  Rather than focusing on *outputs*, they focus on *inputs* (conditions). A counterfactual explanation says, what output would have been produced by the model given a different conditions.

Strong assumptions, such as assuming perfect knowledge of a causal model M of the target domain, imply some responsibility on the part of the authors to either argue that (1) such an assumption is reasonable in practice (a hard sell, in this case); or (2) plausible violations of the assumption do not invalidate the conclusions of the analysis based on the assumptions. In this paper, we have neither. Greater discussion around this point would improve the paper.

The paper is reasonably critical of current work on XAI (e.g., “…the literature on XAI has failed to take up the most relevant lesson for XAI that the literature on causation has to offer…”). When leveling such criticisms, it is important to accurately characterize the goals of the work being critiqued. In at least some portions, the current paper does not accurately characterize the work. Specifically, it assumes that all work on XAI intends to explain why a given action is expected to achieve a particular outcome (how the world works) rather than to explain why a given model recommended a given action (how the model works). This difference is a fundamental one, and it is one that has been discussed extensively among (at least some) researchers working on XAI.

Of course, a quite plausible question is whether it is possible to produce counterfactual explanations (about the model alone) that will be correctly interpreted. The explanation “Our model for credit scoring would have produced a different outcome if your education level had been higher” appears only slightly different than saying “You would be more worthy of credit if you had more education”. The two statements are miles apart in terms of the knowledge required of the world to make them with confidence, but users of an explanation system might easily confuse them.  Furthermore, one might argue (and the current paper does) that counterfactual explanations of the model alone are really not possible without implicit reference to the world. This argument should be foregrounded rather than remain in its current (somewhat buried) position in the paper.

The discussion at the beginning of Section 5 is an extremely important one for the XAI enterprise, and the paper would be improved if the introduction of the paper did a better job of forecasting this future discussion rather than (as it does now) assuming that most XAI literature is doing far more than what Section 5 calls “the first interpretation”.

**Summary:**

A useful theoretical analysis of causal concepts for explainable AI that could use substantial revisions to the front of the paper

---

> ### Author Response · Authors · 2021-11-29
> **Response to reviewer**
>
> Thanks a lot for your positive and detailed feedback! The numbers indicate which paragraph of the review is being addressed.
>
> 1:
> I agree entirely that it is crucial to distinguish between explanations of the model, and explanations of the system in the world that we are modeling, and this is why I do so in the second and third paragraphs. I also agree that most of XAI is focused on the former, and I will make this more explicit in the introduction. However, I am not so sure that most people who are reading work on XAI are aware of this, and I do believe that this in part due to the fact that most work on XAI fails to clearly specify the different types and purposes that an explanation can serve. (See note 5 and 6 below.)
>
> 3:
> I agree that the current formulation fails to take into account the conditions (input). Perhaps this formulation is better: “If one is interested in finding an action that changes already observed conditions to produce a change in an already observed output, then Counterfactual Explanations are called for.” I will reverse the order: first introduce sufficient explanations, then counterfactual explanations.
>
> 4:
> I agree that there is much more to be said about this, but I do want to point out two important elements that are already mentioned in the paper. First, there is definitely a strongly increasing interest in learning partial causal models that are designed specifically to aid and interact with ML-models, which is referenced at the beginning of page 2. Although these are just a few papers, they are all very recent, and as the current conference illustrates it is to be expected that more will follow soon. Second, the paragraph before Definition 3 (briefly) explains how the current deterministic definitions could be generalized to probabilistic definitions, and those would also be applicable to partial causal models. Of course the proof of the pudding is in the eating, and thus I agree that until such a generalization is produced the reader might remain somewhat sceptical. Still, I believe it is important not to conflate conceptual and definitional issues (which are the target of the paper) with practical empirical issues. This paper describes what it *means* to offer (counterfactual, sufficient, etc.) explanations which are action-guiding in the ideal setting of perfect knowledge, and therefore offers a starting point on which to build definitions in epistemically limited settings.
>
> 5 and 6:
> As mentioned in note 1 above, I do believe that the paper carefully distinguishes between action-guiding explanations and explanations of the ML-model itself, but I agree that the paper could also be more explicit about this distinction when levelling criticism at current work in XAI (the quotation of the reviewer offers a good case in point). However, speaking from personal experience I have come across many people who are not themselves experts in the field but would like to make use of its results (such as medical practitioners) who are not at all clear about this distinction. This latter point is very congenial to the reviewer’s point about interpreting counterfactual explanations, and I agree that it deserves more attention. I will therefore add a paragraph to the introduction clarifying that although most experts in XAI are (usually) careful not to conflate the different types of explanations, this does not mean that the same holds for those hearing and using such explanations. This is precisely why it is important to define explanations in such a manner that these ambiguities cannot arise.
>
> 7:
> I am glad to hear that the reviewer values the discussion in Section 5. I agree that, given the importance of counterfactual explanations in XAI, the distinctions made should be mentioned more explicitly as novel contributions of the paper in the introduction, and will adjust this.

---

### Decision · Program_Chairs · 2022-01-12

**Decision:**

Accept (Oral)

**Comment:**

This paper draws on the actual causation literature, and defines and compares several concepts of causal explanation that are arguably relevant to explainable AI. The paper is clearly written and technically sound. The particularly interesting and original contributions of this paper include making explicit the action-guiding implications of the various conceptions of causation and explanation, and relating the notion of actual causation (as opposed to counterfactual explanation) to the important issue of fairness.

On the other hand, this paper works with deterministic rather than probabilistic causal models, which in my view is its main limitation. It is unclear whether generalisations to probabilistic causal models would be as straightforward as the author seems to suggest. (One question is whether one could define a probabilistic notion of actual causation as opposed to a notion of the probability of actual causation, and if so, whether they suggest significantly different notions of explanation.)

Overall I agree with the three reviewers that this is a good paper for the conference.